# Diversity and Effectivity of Indigenous *Mesorhizobium* Strains for Chickpea (*Cicer arietinum* L.) in Myanmar

**Khin Myat Soe [1],\*, Aung Zaw Htwe [2] , Kyi Moe [2] , Abiko Tomomi [3] and Takeo Yamakawa [1]**

[1]  Laboratory of Plant Nutrition, Department of Bioresource and Bioenvironmental Sciences, Faculty of Agriculture, Kyushu University, 744 Motooka, Nishi-ku, Fukuoka 819-0395, Japan; yamakawa@agr.kyushu-u.ac.jp

[2]  Department of Agronomy, Yezin Agricultural University, Yezin 15013, Myanmar; aungzawhtwe333@gmail.com (A.Z.H.); kyimoeyau@gmail.com (K.M.)

[3]  Laboratory of Agroecology, Department of Bioresource Sciences, Faculty of Agriculture, Kyushu University, 111 Harumachi, Kasuya-machi, Kasuya-gun, Fukuoka 811-2307, Japan; abikotomomi@farm.kyushu-u.ac.jp

\*  Correspondence: khinmyatsoe@gmail.com; Tel.: +81-80-5188-1593

**Abstract:** Chickpea (*Cicer arietinum* L.) is one of the world's main leguminous crops that provide chief source of food for humans. In the present study, we characterized thirty isolates of indigenous chickpea rhizobia from Myanmar based on the sequence analysis of the bacterial 16S rRNA gene. The sequence analysis confirmed that all isolates were categorized and identified as the genus *Mesorhizobium* and they were conspecific with *M. plurifarium*, *M. muliense*, *M. tianshanense*, and *M.* sp. This is the first report describing *M. muliense, M. tianshanense,* and *M. plurifurium* from different geographical distribution of indigenous mesorhizobia of chickpea in Myanmar. In order to substitute the use of chemical fertilizers in legume production, there is a need for the production of Biofertilizers with rhizobial inoculants. The effectiveness of Myanmar *Mesorhizobim* strains isolated from soil samples of major chickpea growing areas of Myanmar for plant growth and nitrogen fixation were studied in pot experiments. The nodule dry weight and acetylene reduction activity of the plant inoculated with *Mesorhizobium tianshanense* SalCP19 was significantly higher than the other tested isolates in Yezin-4 chickpea variety. But, *Mesorhizobium* sp. SalCP17 was showed high level of acetylene reduction activity per plant in Yezin-6 chickpea variety.

**Keywords:** chickpea; Myanmar; 16S rRNA gene; *Mesorhizobium*; nitrogen fixation

## 1. Introduction

Grain legumes play an important nutritional role in the diet of millions of people [1]. Among these grain legumes, chickpea (*Cicer arientinum* L.) is one of the most popular and earliest-cultivated legume crop, and seeds are rich in protein content [2]. In Myanmar, it is an important cash crop with increasing demand for its domestic consumption and export. During 2013–2017, chickpea growing areas in Myanmar was about 3.75 million hectares with a production of 2.79 million tons, and average yield was 1489 kg ha$^{-1}$ [3]. Myanmar is the third world chickpea-producer in 2017 in terms of tons produced and value ($).

Nitrogen (N) fixation through legume-*Rhizobium* symbiosis is important for enhancing agricultural productivity and is therefore of great economic interest [4]. Leguminous crops have a reputation for maintaining soil fertility since it can assimilate nitrogen from the atmosphere through symbiotic biological N$_2$ fixation (BNF) with Rhizobia [5]. Biological nitrogen fixation (BNF) is an important component of sustainable agriculture [6], and rhizobial inoculants have been applied frequently as biofertilizers.

Urea is the main source of nitrogen applied to all crops grown in Myanmar but it is very expensive [7]. Commercial production of Rhizobium inoculant has been developed for several decades. The main objective of using Rhizobium inoculant is to substitute the nitrogenous fertilizers in food legume production. It is cheaper and lighter in weight than urea and easier for the farmers [8]. To improve high quality inoculant production, the evaluation of highly effective rhizobial strains for specific legume is one of the principle obligations [9,10]. Indigenous rhizobial strains also play an important role since they have adapted to local environmental conditions. Therefore, the investigation of effective indigenous rhizobial strains should be considered for current inoculant production in Myanmar.

Last 10 years ago, Than [11] collected 53 native Mesorhizobial isolates from 11 Townships of Sagaing Region in Myanmar and four effective isolates were screened out based on the abilities to improve nodule numbers, nodule dry weights, and shoot dry weights of chickpea plants after sowing under by using modified Leonard's bottle jar method in the screen-house of Department of Plant Pathology, Yezin Agricultural University, Myanmar. They were also determined the specific *Mesorhizobium ciceri* isolate was analyzed by using Random Amplified Polymorphic DNA (RAPD) technique. However, further study on the effects of other native Mesorhizobia isolates for other chickpea cultivars from Myanmar on the determination of their effectiveness on plant growth and $N_2$ fixation are needed in order to guarantee the potential of these indigenous microbes for chickpea production in Myanmar. Moreover, it is necessary to characterize and identify the nitrogen-fixing bacteria in order to substitute the utilization of urea fertilizer in food legume crops.

For this reason, we aimed to isolate indigenous root nodule bacteria from collected soil samples of major chickpea growing areas of Myanmar, to identify the phylogenetic diversity of indigenous chickpea-nodulating Mesorhizobia in Myanmar based on sequence analysis of the 16S rRNA region of the isolates and, to select the effective indigenous Myanmar *Mesorhizobium* strains for plant growth and nitrogen fixation of Myanmar chickpea varieties is necessary for investigation.

## 2. Materials and Methods

### 2.1. Analysis of Collected Soil Samples

Total six collected soil samples from major chickpea growing areas of Myanmar were analyzed at Plant Nutrition Laboratory, Faculty of Agriculture, Kyushu University, Fukuoka, Japan. For each collected soil sample, soil pH $H_2O$ (1:2.5 soil:$H_2O$) was measured using a pH meter (HM-10P; DKK-TOA Corp., Tokyo, Japan). Soil samples were also digested using the salicylic acid–sulfuric acid–hydrogen peroxide method [12]; then, total N was examined using the indophenol method [13], and total P was tested using the ascorbic acid method [14]. Total K was analyzed using an atomic absorption spectrophotometer (Z-5300; Hitachi, Tokyo, Japan) afterward samples were digested. To analyze mineralizable N, we firstly used the soil incubation method [15] followed by the indophenol method [13].

### 2.2. Isolation of Indigenous Root Nodule Bacteria from Soil Samples of Major Chickpea (Cicer Arietinum) Growing Areas of Myanmar

One gram of each composite soil sample was diluted with 99 mL of sterilized one-half strength modified Hoagland nutrient solution (MHN) of pH 6.5 in a 200 mL conical flask. The MHN contain the following chemicals; 8.9 g of NaFe EDTA was dissolved in deionized water and filled up to 500 mL. 183.79 g of $CaCl_2.2H_2O$ was dissolved in deionized water and filled up to 2 L. 108.9 g of $K_2SO_4$, 34.25 g of $KH_2PO_4$, 126.65 g of $MgSO_4.H_2O$, 55.5 mg of $ZnSO_4.7H_2O$, 390.5 mg of $MnSO_4.H_2O$, 20.6 mg of $CuSO_4.5H_2O$, 725 mg of $H_3BO_3$ and 4.63 mg of $MoO_3.2H_2O$ were dissolved in deionized water and filled up to 2 L [16,17].

The flasks were shaken on a rotary shaker at 120 rpm for one hour to prepare a well-mixed soil suspension. The culture pots (1 L volume) were filled with 1 L of vermiculite and 0.6 L of MHN nutrient solution. The pots were covered with aluminum foil and autoclaved at 121 °C for 20 min. For

surface sterilization, the seeds were soaked in a 2.5% sodium hypochlorite solution for 5 min, rinsed five times with 10 mL of 99.5% ethanol and washed five times with sterilized MHN nutrient solution to remove traces of sodium hypochlorite and ethanol. Yezin-4 chickpea variety, common use in Myanmar was used as trap hosts for all soil samples.

The six culture pots using chickpea Yezin-4 variety with six soil suspensions and control pot were prepared. Some good seeds were surface-sterilized and six sterilized seeds were planted in the sterilized vermiculite pots. 5 mL aliquot of soil suspension was inoculated per seed. The control was planted without inoculation to assess the possibility of contamination. The plants were cultivated in the incubator (25 °C and 16 h light) for four weeks. Autoclaved deionized water was poured when the original weight of the pots decreased by around 300 g.

After carefully uprooting, the five nodules ($\geq$ 2 mm in diameter) were collected per pot. For surface sterilization, the nodules were soaked in 70% ethanol for 3 min, 2.5% sodium hypochlorite (NaClO) solution for 15 min and washed five times with 0.9% autoclaved sodium chloride (NaCl) solution. The surface sterilized nodules were transferred separately into the autoclaved small test tubes and crushed. For every sample, a loopful of the suspension was streaked on Yeast Extract Mannitol Agar (YMA) plates [18] containing 25-µg Congo red [19]. The plates were incubated at 30 °C for 3–5 days for fast growing bacteria. Finally, purified 30 indigenous root nodule bacteria isolates were isolated from soil samples of major Chickpea (*Cicer arietinum*) growing areas of Myanmar. The collected isolates were nominated as DemCP1 to KenCP30.

### 2.3. DNA Extraction, PCR Analysis, and Phylogenetic Analysis

For DNA extraction, the collected isolated were streaked onto A1E agar plates and incubated at 30 °C for 7 days [20]. A single pure colony of each isolate from A1E plates was cultured in AIE liquid medium at 30 °C for 5 days to obtain the required optimum density ($0.4 < OD_{600nm} < 0.6$). Total DNA was extracted using ISOPLANT (Nippon gene, Tokyo, Japan), following instructions from the manufacturer. The DNA concentrations were calculated using NIH Image 1.62 (National Institutes of Health, Bethesda, MD, USA) after agarose gel electrophoresis (0.3% agarose gel in 1 TAE buffer), staining with ethidium bromide (Toyobo, Tokyo, Japan), and destaining in 1 TAE buffer.

The primers 16S-F (5′-AGAGTTTGATCCTGGCTCAG-3′) and 16S-R2 (5′-CGGCTACCTTGTTA CGACTT-3′) were used to amplify the 16S rRNA region of mesorhizobia. The PCR reaction consisted of a pre-run at 94 °C for 5 min, denaturation at 94 °C for 30 s, annealing at 60 °C for 30 s, and extension at 72 °C for 1 min. The cycle was repeated for 33 cycles, followed by a final extension at 72 °C for 10 min [21]. PCR products were purified using the Wizard Gel and PCR Clean-up System (Promega, Madison, WI, USA). Purified PCR products ($\geq$50 ng µL$^{-1}$) were subjected to direct sequencing by Macrogen (Tokyo, Japan), using the primer set described above. Raw sequence results were edited using MEGA version 6 software [22] to create 16S sequence fragments.

For homology searches, sequences were compared with the DNA Data Bank of Japan (DDBJ) using the Basic Local Alignment Search Tool (BLAST) program [23]. To construct the phylogenetic tree, sequences of type strains and closely related strains of *Mesorhizobium* genospecies were retrieved from the BLAST database. All selected sequences including type strains and closet strains were aligned using the CLUSTALW function of the MEGA version 6 software [22]. After alignment, a phylogenetic tree was constructed according to the neighbor-joining method [24]. The phylogenetic tree was bootstrapped with 1000 replications of each sequence to evaluate the tree topology for reliability. Genetic distances were calculated using the Kimura two-parameter model [25].

### 2.4. Nucleotide Sequence Accession Numbers

The nucleotide sequences of 16S rRNA genes of 30 isolates were deposited in the DDBJ under the set of accession numbers LC515471 to LC515500.



### 2.5. Myanmar Chickpea Cultivars and Mesorhizobium Strains

Myanmar Chickpea (*Cicer arietinum*) Cultivars, Yezin-4 and Yezin-6 were collected from Food Legumes Section, Department of Agricultural Research, Yezin, Myanmar. The Chickpea Yezin-4, most widely grown cultivar in Myanmar, was used for screening of the 30 indigenous *Mesorhizobium* strains. For effectiveness of selected *Mesorhizboium* strains, Chickpea Yezin-4 and Yezin-6 cultivars from Myanmar were investigated. The purified thirty *Mesorhizobium* strains were cultured in A1E liquid media [26] on a rotary shaker (100 rpm) at 30 °C for 7 days. The cultures were diluted with sterilized N-free half-strength modified Hoagland nutrient (MHN) solution [27] to obtain about $10^7$ cells mL$^{-1}$ for *Mesorhizobium* strains.

### 2.6. Screening the Effectiveness of Mesorhizobial Strains for Nitrogen Fixation on Yezin-4 chickpea Variety

The purified thirty indigenous mesorhizobial strains were screened for nitrogen fixing on Yezin-4 chickpea variety was investigate by using plastic pots with vermiculite and MHN solution. The pot experiment was conducted 16 December, 2018 to 13 January, 2019 in Plant Nutrition Laboratory, Kyushu University, Japan. Some Yezin-4 chickpea seeds were surface sterilized [28] and germinated on sterile petri dishes with filter paper. The six germinated seeds were sown into the sterilized pot with vermiculite. The non-inoculated treatment, a control treatment was also provided. The weights of original pots were measured. The plants were cultivated in incubator (25 °C and 70% relative humidity). During the growing period, sterilized water was irrigated. The plants in each pot were uprooted and carefully washed with water so as not to detach the nodules. The acetylene reduction assay (ARA) was performed according to Haider et al., [29] to measure nitrogenase activity. The green gram plants were cut at the cotyledonary nodes. Then, the green gram roots with intact nodules was placed in a 100 mL conical flask and sealed with a serum stopper. A 12 mL aliquot of acetylene ($C_2H_2$) gas was injected into the flask to replace the air with acetylene. The flasks containing roots with intact nodules were incubated at room temperature and 1.0 mL subsamples were analyzed at 5 and 65 min, respectively. The ARA value, in terms of ethylene ($C_2H_4$) production per plant, was measured using a flame ionization gas chromatograph (GC-14A, Shimadzu, Kyoto, Japan) equipped with a stainless steel column (3 mm diameter, 0.5 m length). The column was filled with Porapak R 60–80 mesh (Nicalai Tesque, Inc., Kyoto Japan). Column, injection and detection temperatures were 35, 45, and 170 °C, respectively. $N_2$ gas was used as the carrier gas at a flow rate of 45 mL min$^{-1}$. The number of nodules was counted after the assay. Shoots, roots, and nodules were collected separately and oven dried at 70 °C for 48 h. The data was recorded for dry weight determination.

### 2.7. Evaluation the Effectiveness of Selected Mesorhizboium Strains on Two Myanmar Chickpea Cultivars: Yezin-4 and Yezin-6

The five *Mesorhizobium* strains, *M.* sp. DemCP4, *M. muliense* MgnCP6, *M. plurifurium* MinCP15, *M.* sp. SalCP17, *M. tianshanense* SalCP19 were selected based on the results of the above screening experiment in their potential efficiency on ARA per plant. The pot experiment was performed in completely randomized design with three replicates during 30 July, 2019 to 27 August, 2019 in Plant Nutrition Laboratory, Kyushu University, Japan. The inoculation and growing condition of pot experiment was also conducted as the above experiment. ARA per plant, nodule, root, and shoot dry weight were determined after four weeks.

### 2.8. Statistical Analysis

Data were analyzed using the STATISTIX 8 software (Analytical Software, Tallahassee, FL, USA), and treatment means were compared by Tukey's HSD test ($p < 0.05$) for the collected parameters.

## 3. Results

### 3.1. Diversity of Indigenous Mesorhizobium Strains for Myanmar Chickpea (Cicer arietinum L.) Cultivars

The thirty root nodule bacteria were isolated from Myanmar chickpea host of six different major chickpea growing areas of Myanmar (Tables 1 and 2 and Figure 1). These strains were proved as pure *Mesorhizobium* strains on YMA plates after 3–5 days (fast-growers) incubation [19]. In YMA plates, the mesorhizobial colonies reached 1–3 mm diameter with undulated pulvinate and entirely pulvinate shapes (Table 3).

Neighbor-joining trees for each gene had similar overall tree topologies. Groups were selected on the basis of the minimum standard changes between named species in the 16S rRNA phylogram (Figure 2), and all groups were well supported in neighbor-joining analyses which had less than 50% bootstrap support in the neighbor-joining tree. The results of the phylogenetic analysis based on the 16S sequence, indicated that all the 30 isolates belonged to the genus *Mesorhizobium*.

**Table 1.** Soil classification, location and climate of soil samples collected from major chickpea growing areas of Myanmar.

| Soil Sampling Site | Soil Classification * | Location | Climate ** (Temp; RF) |
|---|---|---|---|
| Demoso, Kayah State | Mountainous Brown Forest Soil | 19°40′ N 97°12′ E | 17-29 °C, 1045 mm |
| Myingyan, Mandalay Region | Dark Compact Soil | 21°27′ N 95°23′ E | 21-34 °C, 646 mm |
| Minbu, Magway Region | Meadow and Meadow Alluvial Soil | 21°10′ N 94°52′ E | 21-34 °C, 767 mm |
| Salingyi, Sagaing Region | Meadow and Meadow Alluvial Soil | 21°58′ N 95°05′ E | 22-33 °C, 803 mm |
| Pyinmanar, Nay Pyi Taw Region | Meadow Alluvial Soil | 19°45′ N 96°12′ E | 21-33 °C, 1302 mm |
| Kengtung, Eastern Shan State | Red Earth and Yellow Earth | 21°17′ N 99°36′ E | 17-29 °C, 1346 mm |

Sources: * Shein, H.A. The soil types and characteristics of Myanmar. Department of Agriculture, Ministry of Agriculture, Livestocks and Irrigation: Nay Pyi Taw, Myanmar, 2015. ** Aung, L.L.; Zin, E.E.; Theingi, P.; Elvera, N.; Aung, P.P.; Han, T.T.; Oo, Y.; Skaland, R.G. Myanmar Climate Report published by Department of Meteorology and Hydrology Myanmar, Ministry of Transport and Communications, Government of the Republic of the Union of Myanmar, 2017.

**Table 2.** Physicochemical properties of soil.

| Physicochemical Property | Demoso, KYS | Myingyan, MDR | Minbu, MGR | Salingyi, SGR | Pyinmanar, NPTR | Kengtung, ESS |
|---|---|---|---|---|---|---|
| Soil pH (Soil:$H_2O$; 1:2.5) | 7.15 | 7.44 | 6.30 | 7.27 | 6.15 | 4.79 |
| Total N (%) | 0.17 | 0.06 | 0.19 | 0.06 | 0.18 | 0.16 |
| Mineralizable N (g/kg) | 1.43 | 0.32 | 1.56 | 0.48 | 1.81 | 0.51 |
| Total $P_2O_5$ (%) | 0.12 | 0.02 | 0.12 | 0.03 | 0.31 | 0.10 |
| Total K2O (%) | 1.08 | 0.18 | 1.44 | 0.41 | 1.20 | 0.77 |

KYS: Kayah State, MDR: Mandaly Region, MGR: Magway Region, SGR: Sagaing Region, NPTR: Nay Pyi Taw Region, ESS: Eastern Shan State. Soil sample analyses were performed in the Plant Nutrition Laboratory, Faculty of Agriculture, Kyushu University, Japan.

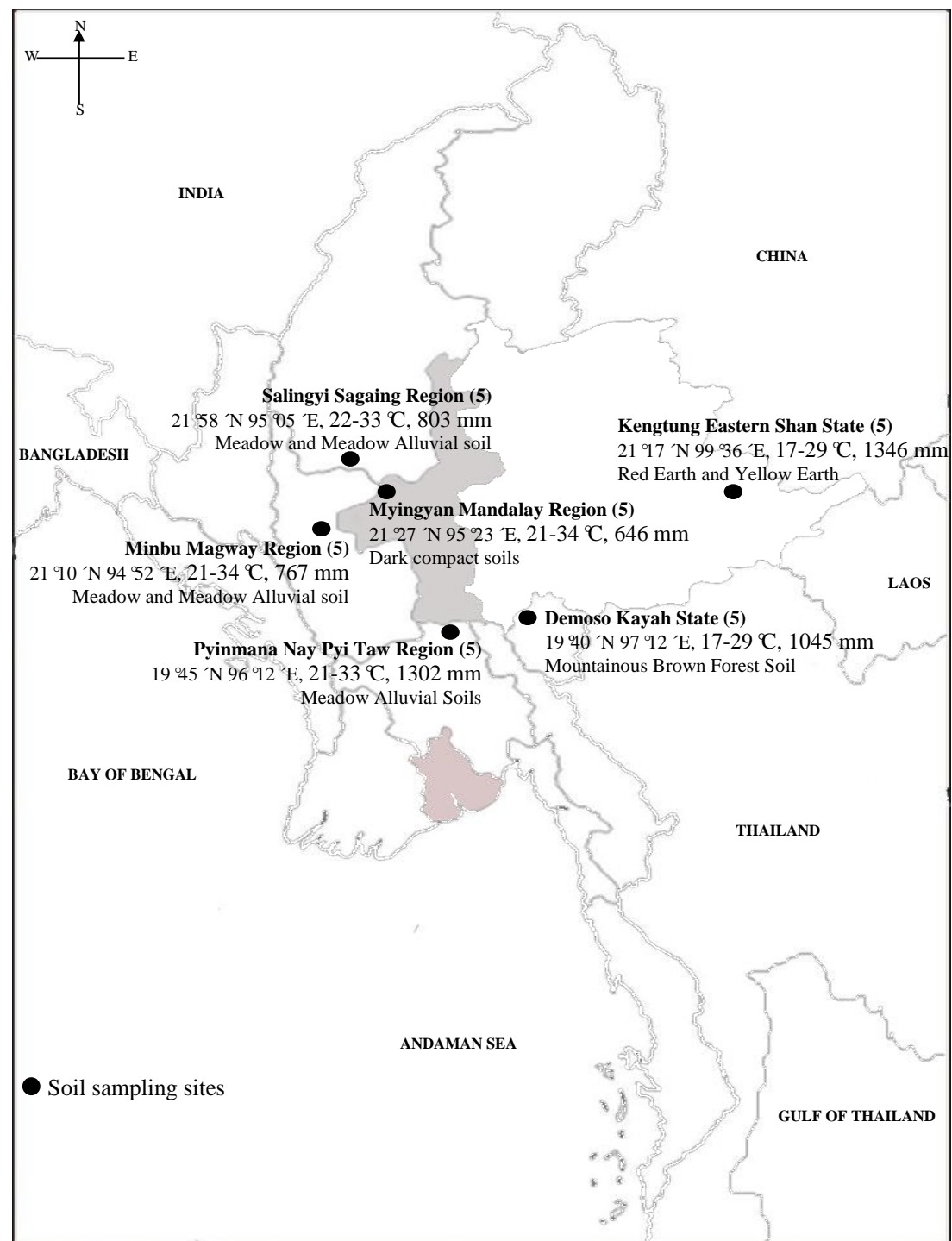

**Figure 1.** Map of Myanmar showing soil sampling sites from chickpea major growing areas in Myanmar. Numbers in parentheses represent the total number of isolates collected from each sites.

**Table 3.** The origin and morphological characteristics of Chickpea mesorhizobial isolates from Myanmar on yeast mannitol agar (YMA) with congo red plates after 3-5 days incubation at 30 ∘C.

| Strain Name | Genus and Species | Isolated Origin | Dendrogram Cluster | Shape | Size (mm) |
|---|---|---|---|---|---|
| DemCP1 | *Mesorhizobium* sp. | Demoso, Kayah State | Ms1 | UF | 2.0 |
| DemCP2 | *Mesorhizobium* sp. | Demoso, Kayah State | Ms2 | UF | 2.0 |
| DemCP3 | *Mesorhizobium* sp. | Demoso, Kayah State | Ms2 | UF | 2.0 |
| DemCP4 | *Mesorhizobium* sp. | Demoso, Kayah State | Ms2 | UF | 2.0 |
| DemCP5 | *Mesorhizobium* sp. | Demoso, Kayah State | Ms2 | UF | 2.0 |
| MgnCP6 | *Mesorhizobium muleiense* | Myingyan, Mandalay Region | Mm1 | UF | 1.5 |
| MgnCP7 | *Mesorhizobium muleiense* | Myingyan, Mandalay Region | Mm1 | UF | 1.5 |
| MgnCP8 | *Mesorhizobium tianshanense* | Myingyan, Mandalay Region | Mt1 | UF | 1.5 |
| MgnCP9 | *Mesorhizobium muleiense* | Myingyan, Mandalay Region | Mm1 | UF | 1.5 |
| MgnCP10 | *Mesorhizobium muleiense* | Myingyan, Mandalay Region | Mm1 | UF | 1.5 |
| MinCP11 | *Mesorhizobium plurifarium* | Minbu, Magway Region | Mp1 | EP | 3.0 |
| MinCP12 | *Mesorhizobium plurifarium* | Minbu, Magway Region | Mp3 | EP | 3.0 |
| MinCP13 | *Mesorhizobium plurifarium* | Minbu, Magway Region | Mp3 | EP | 3.0 |
| MinCP14 | *Mesorhizobium plurifarium* | Minbu, Magway Region | Mp3 | EP | 3.0 |
| MinCP15 | *Mesorhizobium plurifarium* | Minbu, Magway Region | Mp3 | EP | 3.0 |
| SalCP16 | *Mesorhizobium tianshanense* | Salingyi, Sagaing Region | Mt1 | UF | 2.0 |
| SalCP17 | *Mesorhizobium* sp. | Salingyi, Sagaing Region | Ms2 | UF | 2.0 |
| SalCP18 | *Mesorhizobium tianshanense* | Salingyi, Sagaing Region | Mt1 | UF | 2.0 |
| SalCP19 | *Mesorhizobium tianshanense* | Salingyi, Sagaing Region | Mt1 | UF | 2.0 |
| SalCP20 | *Mesorhizobium tianshanense* | Salingyi, Sagaing Region | Mt1 | UF | 2.0 |
| PyiCP21 | *Mesorhizobium plurifarium* | Pyinmanar, Nay Pyi Taw Region | Mp1 | EP | 2.5 |
| PyiCP22 | *Mesorhizobium plurifarium* | Pyinmanar, Nay Pyi Taw Region | Mp1 | EP | 2.5 |
| PyiCP23 | *Mesorhizobium plurifarium* | Pyinmanar, Nay Pyi Taw Region | Mp2 | EP | 2.5 |
| PyiCP24 | *Mesorhizobium plurifarium* | Pyinmanar, Nay Pyi Taw Region | Mp1 | EP | 2.5 |
| PyiCP25 | *Mesorhizobium plurifarium* | Pyinmanar, Nay Pyi Taw Region | Mp1 | EP | 2.5 |
| KenCP26 | *Mesorhizobium plurifarium* | Kengtung, Eastern Shan State | Mp4 | UF | 2.0 |
| KenCP27 | *Mesorhizobium plurifarium* | Kengtung, Eastern Shan State | Mp1 | UF | 2.0 |
| KenCP28 | *Mesorhizobium plurifarium* | Kengtung, Eastern Shan State | Mp1 | UF | 2.0 |
| KenCP29 | *Mesorhizobium plurifarium* | Kengtung, Eastern Shan State | Mp2 | UF | 2.0 |
| KenCP30 | *Mesorhizobium plurifarium* | Kengtung, Eastern Shan State | Mp1 | UF | 2.0 |

Mp1 = Mesorhizobium plurifarium cluster 1; Mp2 = Mesorhizobium plurifarium cluster 2; Mp3 = Mesorhizobium plurifarium cluster 3; Mp4 = Mesorhizobium plurifarium cluster 4; Ms1 = Mesorhizobium sp.cluster 1; Ms2 = Mesorhizobium sp.cluster 2; Mm1 = Mesorhizobium muleiense cluster 1; Mt1 = Mesorhizobium tianshanense cluster 1; UP = Undulated-pulvinate, EP = Entirely-pulvinate.

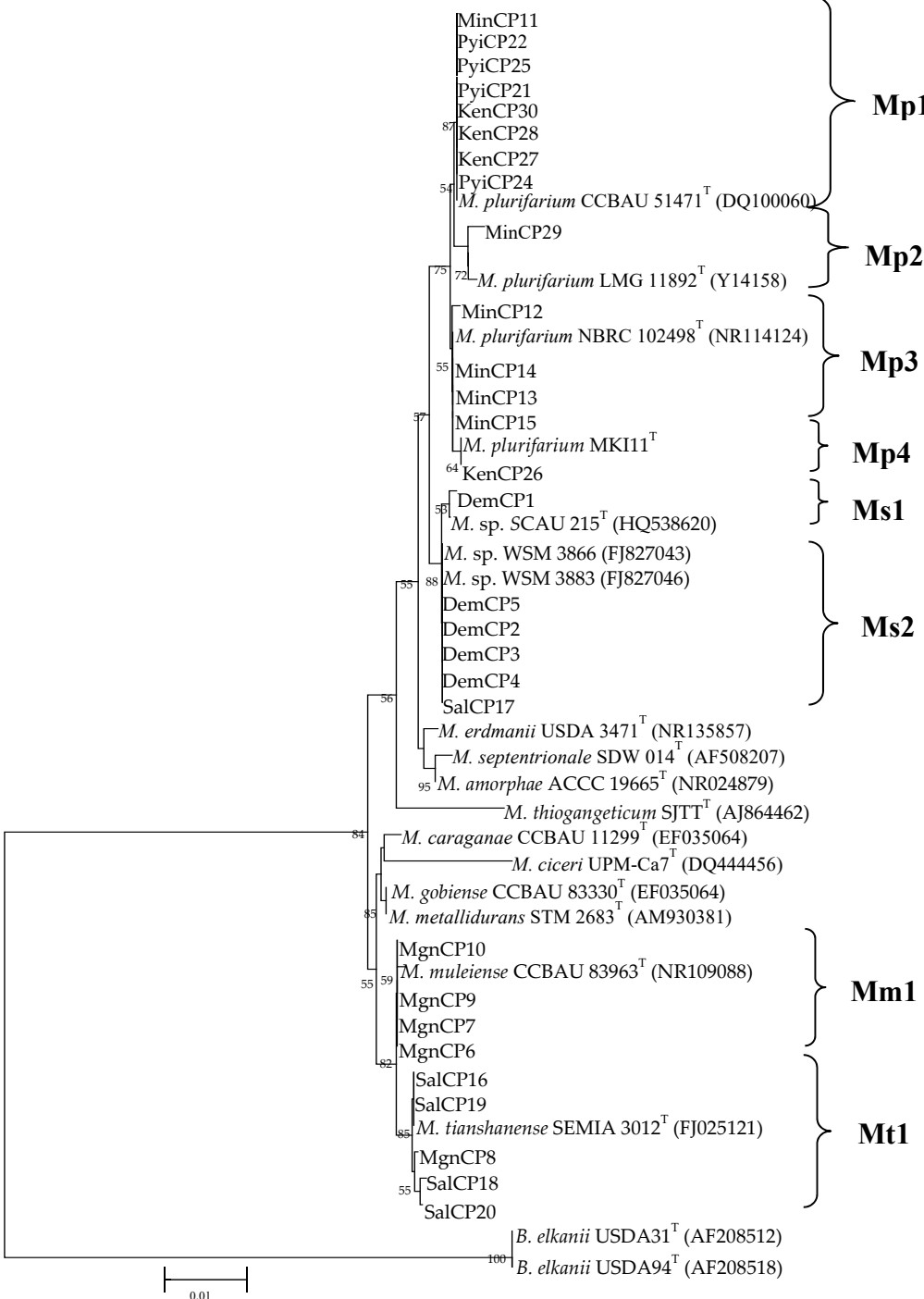

**Figure 2.** Position of the 30 strains in the phylogenetic tree based on the 16S rRNA sequences of related *Mesorhizobium* strains (in italics) retrieved from GenBank. The tree was constructed by the neighbor-joining method with the Kimura 2-parameter (K2P) distance correlation model and 1,000 bootstrap replications. Bootstrap values above 50% are indicated at the nodes. Bar, 0.02 $K_{nuc}$ in nucleotide sequences. Accession numbers of the reference strains, including all type strains of *Mesorhizobium*, are shown in parentheses. *M: Mesorhizobium* and *B: Bradyrhizobium*. The clustering of isolates and their distribution throughout the total studied area is noted into the tree: Mp1 = *Mesorhizobium plurifarium* cluster 1; Mp2 = *Mesorhizobium plurifarium* cluster 2; Mp3 = *Mesorhizobium plurifarium* cluster 3; Mp4 = *Mesorhizobium plurifarium* cluster 4; Ms1 = *Mesorhizobium* sp.cluster 1; Ms2 = *Mesorhizobium* sp.cluster 2; Mm1 = *Mesorhizobium muleiense* cluster 1; Mt1 = *Mesorhizobium tianshanense* cluster 1.

The eight clusters were identified in the phylogenetic tree including four clusters of *M. pulrifarium* (Mp1, Mp2, Mp3, and Mp4), two clusters of M. sp. (Ms1 and Ms2), one cluster of each *M. muleiense* (Mm1) and *M. tianshanense* (Mt1) (Figure 2). Among these clusters, two clusters of Mp1 and Mp3 were 99% sequence similarity with *M. pulrifarium* CCBAU 51471[T] and *M. pulrifarium* NBRC 172498[T] while two clusters of Mp2 and Mp4 were 98% sequence similar to *M. pulrifarium* LMG 11892[T]. Among two clusters belonging to *M.* sp., Ms1 and Ms2 were showing 99% and 98% sequence similarity with *M.* sp. SCAU 215[T] and *M.* sp. WSW 3883[T], respectively. The last two clusters of Mm1 and Mt1 groups were related to *M. muleiense* CCBAU 83963[T] and *M. tianshanense* SEMIA 3012[T] with at least 98% and 97% sequence similarity.

Among the collected strains, almost all strains from Minbu Magway Region, Pyinmanar Nay Pyi Taw Region, and Kengtung Eastern Shan State chickpea growing regions were identified as *M. pulrifarium strains*. At Mingyan Mandalay Region, most of the strains were identified as *M. muleiense* except MgnCP8 which belonged to *M. tianshanense*. Most of the strains from Salingyi Sagaing Region were related to *M. tianshanense* and only one strain was *M.* sp. (Table 3 and Figure 2). However, almost all the *M.* sp. strains were found in Demoso Kayah State chickpea growing region.

In this study, cluster Ms1 and Ms2 were observed only in Demoso Kayah State chickpea growing area. In Myingyan Mandalay Region, cluster Mm1 and Mt1 were found but in Salingyi Sagian Region, cluster Mt1 and Ms2 were distributed. However, Mp1 cluster was distributed evenly in Minbu, Magway Region, Pyinmana Mandalay Region and Kengtung Eastern Shan State chickpea growing regions. The percent distribution of isolates in the different sites is shown in (Table 4).

**Table 4.** Percentage distribution (%) of isolates at each field site.

| Field Sites | *M.plurifarium* | | *M.* sp. | | *M. muleiense* | | *M. tianshanense* | |
|---|---|---|---|---|---|---|---|---|
| | % | No. Clusters | % | No. Clusters | % | No. Clusters | % | No. Clusters |
| Demoso, Kayah State | - | - | 100 | 2 | - | - | - | - |
| Myingyan, Mandalay Region | - | - | - | - | 80 | 1 | 20 | 1 |
| Minbu, Magway Region | 100 | 2 | - | - | - | - | - | - |
| Salingyi, Sagaing Region | - | - | 20 | 1 | | - | 80 | 1 |
| Pyinmanar, Nay Pyi Taw Region | 100 | 2 | - | - | - | - | - | - |
| Kengtung, Eastern Shan State | 100 | 3 | - | - | - | - | - | - |
| Total | 50 | | 20 | | 13 | | 17 | |

Total percentage distribution (%) of isolates in the total studied area. *M.: Mesorhizobium.*

### 3.2. Screening of Effective Bacterial Strains by Yezin-6 for Nitrogen Fixation

In the screening experiment, the effective strains were determined their potential ability in the N fixation analyzed by means of ARA per plant. Each bacterial strain that responded on chickpea Yezin-4 was expressed in Figure 3. The higher ARA per plant was found in plants inoculated by MgaCP6, MinCP15, SalCP17, and SalCP19. From this experiment, five indigenous Bradyrhizibium strains were screened out for further studies based on their effectiveness in ARA per plant. There were four Mesorhizobial strains from the highest ARA per plant value of *M. muleiense* MgaCP6, *M. plurifurium* MinCP15, *M.* sp. SalCP17, and *M. tianshanense* SalCP19 and one strain from the middle ARA per plant value of *M.* sp.DemCP4.

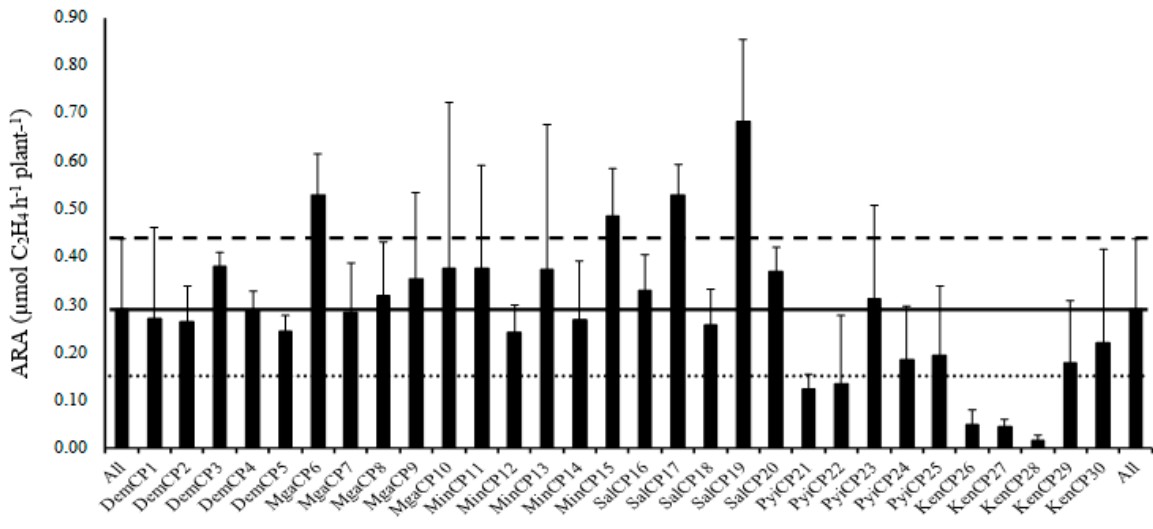

Mesorhizobial isolates

**Figure 3.** Acetylene reduction activity per plant of Yezin4 Chickpea (*Cicer arietinum*) variety inoculated with 30 purified *Mesorhizobium* strains after four weeks, Bars mean standard deviation levels. The middle line indicates average acetylene reduction activity per plant of all strains; the upper line means the higher acetylene reduction activity per plant and the lower line refers the lower acetylene reduction activity per plant of all strains.

### 3.3. Effectivity of Selected Mesorhizobium Strains on Yezin-4 and Yezin-6 of Two Myanmar Chickpea Varieties

Effectiveness of selected Mesorhizobial strains on ARA per plant, nodule, root and shoot dry weight of Yezin-4 and Yezin-6 chickpea varieties in Table 5. The ARA per plant of selected five Mesorhizobial strains was significant affected on Yezin-4 and Yezin-6 chickpea varieties. It was found that significant higher in ARA per plant ($p < 0.05$) was observed by *M.* sp. SalCP17 and *M. tianshanense* SalCP19 than *M.* sp. DemCP4 in Yezin-4 chickpea variety. In Yezin-6 chickpea, the *M. tianshanense* SalCP17 was insignificant differences that compared with *M. muleiense* MgnCP6, *M. plurifurium* MinCP15, and *M.* sp.SalCP19 in ARA per plant in Table.

**Table 5.** Effectiveness of selected Myanmar Mesorhizobial strains on acetylene reduction activity (ARA), nodule, root and shoot dry weight of Yezin4 and Yezin6 Chick Pea varieties after four weeks.

| Treatment | Yezin-4 | | | | Yezin-6 | | | |
|---|---|---|---|---|---|---|---|---|
| | NDW (mg plant$^{-1}$) | RDW (g plant$^{-1}$) | SDW (g plant$^{-1}$) | ARA (µmole $C_2H_4$ h$^{-1}$ plant$^{-1}$) | NDW (mg plant$^{-1}$) | RDW (g plant$^{-1}$) | SDW (g plant$^{-1}$) | ARA (µmole $C_2H_4$ h$^{-1}$ plant$^{-1}$) |
| DemCP4 | 8.77 b | 0.07 a | 0.17 a | 0.31 b | 7.13 b | 0.08 a | 0.15 a | 0.24 b |
| MgnCP6 | 14.57 ab | 0.08 a | 0.21 a | 0.54 ab | 10.27 ab | 0.08 a | 0.19 a | 0.31 ab |
| MinCP15 | 9.87 b | 0.09 a | 0.21 a | 0.46 ab | 9.50 ab | 0.09 a | 0.19 a | 0.39 ab |
| SalCP17 | 14.47 ab | 0.10 a | 0.22 a | 0.55 a | 13.23 a | 0.10 a | 0.22 a | 0.57 a |
| SalCP19 | 22.87 a | 0.09 a | 0.22 a | 0.66 a | 12.83 ab | 0.09 a | 0.20 a | 0.54 ab |

Means in each column followed by different letters differed significantly at $p < 0.05$ (Tukey's test), NDW, RDW, SDW means nodule, root and shoot dry weight per plant and ARA means $C_2H_4$ produced per hour per plant, NDW and ARA of uninoculated treatment is zero for both chickpea varieties, RDW of uninoculated treatment is 0.04 g plant$^{-1}$ for Yezin-4 and 0.05 g plant$^{-1}$ for Yezin-6 chickpea variety; and SDW of uninoculated treatment is 0.04 g plant$^{-1}$ for Yezin-4 and 0.11 g plant$^{-1}$ for Yezin-6 chickpea variety.

On the other hand, the *M. tianshanense* SalCP19 showed the highest nodule dry weight of 22.87 mg plant$^{-1}$ in Yezin-4 but the isolated strain *M.* sp. SalCP17 was nodule dry weight of 13.23 mg plant$^{-1}$ in Yezin-6 chickpea variety. In Yezin-4 variety, inoculation with *M. tianshanense* SalCP17 gave significant higher in nodule dry weight than *M.* sp. DemCP4 and *M. plurifurium* MinCP15. In Yezin-6 variety, *M. tianshanense* SalCP19 gave the highest nodule dry weight among the tested strains and this strain

was significant difference from those given by *M.* sp. DemCP4 since there were no differences about dry shoot weight among the strains in both chickpea varieties. Based on the results of effectiveness experiment, it was clear that all selected Mesorhizobial strains were not effective root dry weight parameter on Yezin-4 and Yezin-6 chickpea varieties.

## 4. Discussion

Chickpea is grown all over the world in about 57 countries under diverse environmental conditions [3]. The symbiotic association between legumes and rhizobia is one of the most important contributors to the world's supply of biologically fixed nitrogen to agriculture. Effective symbiosis can only be achieved when the nodules are formed by effective rhizobia. The symbiotic relationship between rhizobia and chickpea has not been extensively analyzed, and there are few studies addressing the genetic diversity of chickpea rhizobia [30–33].

Sequence analysis of 16S ribosomal RNA (rRNA) has been developed used as one of the most important methods in taxonomy and phylogenic analysis of bacteria [34–37]. In Ethiopia, the collected root nodule bacteria from chickpea were observed the genus *Mesorhizobium* by 16S rRNA gene [38]. These results were dependable with former reports that showed chickpea rhizobia were more closely associated to *Mesorhizobium* species [39,40]. The present study focused on the investigation of 16S rRNA region of 30 Mesorhizobial strains were successfully isolated from the different soil samples of major chickpea growing areas in Myanmar and proved as pure *Mesorhizobium* strains [19].

A phylogenetic analysis of the *Mesorhizobium* genus was a good model to investigate rhizobia genome evolution [2]. The current phylogenetic analysis was related to *M. pulrifurium*, *M. muleiense*, *M. tianshanense* and *M.* sp. The two clusters of Mp1 and Mp3 were 99% sequence similarity with *M. pulrifarium* CCBAU 51471[T] and *M. pulrifarium* NBRC 172498[T] while two clusters of Mp2 and Mp4 were 98% sequence similar to *M. pulrifarium* LMG 11892[T] in Myanmar. In present study, total 15 *M. plurifarium* strains from Mingyan Mandalay Region, Myanmar with a soil pH of 6.30, Pyinmanar Nay Pyi Taw Region, Myanmar with a soil pH of 6.15 and Kengtung Eastern Shan State, Myanmar with a soil pH of 4.79 were isolated from different location of major chickpea growing areas of Myanmar. This is evidence that the five *M. plurifarium* strains formed a cluster at a similarity of 87% Mexico in a numerical taxonomy and these strains were sensitive to salty-alkaline conditions [41] and twenty-one *M. plurifarium* strains were isolated from Korea [42]. Our result pointed out that *M. plurifurium* was widely distributed in the major chickpea growing areas of Myanmar with different soil nutrients and soil pH levels.

This study reports the first findings of *M. muleiense*, *M. plurifurium*, and *M. tianshanense* from soil samples of major chickpea growing areas of Myanmar. There were some reports on isolation and classification of *M. muleiense* has been recently described as the dominant chickpea rhizobia in Xinjiang, China [43,44] was indicated chickpea rhizobia with different biogeographic patterns. Therefore, *M. muleiense* can be considered as the main chickpea rhizobial species specific to alkaline soils in the Northwest of China adapted to this local environment [45]. Recent phylogenetic studies on Mesorhizobial strains from different geographical locations has shown that other species of *Mesorhizobium* may effectively nodulate the legume crops, including *M. tianshanense*, *M. amorphae,* and *M. muleiense* [46–49]. On the other hand, *M. plurifarium* and *M. tianshanense* has been reported in other countries [37,50] were found in the present study.

The results indicated that *Mesorhizobium* strains tolerates a wider range of soil types and pH conditions because it had nearly equal proportions of its strains isolated from the highly acidic and the higher pH. The finding of significant phylogenetic signals for pH indicates that despite the wider tolerance range of *Mesorhizobium* as a genus, the distribution of various strains is phylogenetically structured. Thus, for each of the different soil types and pH conditions of the Mesorhizobial strains collected from soil samples of major chickpea growing areas of Myanmar, there are particular strains of *Mesorhizobium* that are adapted to them. This is consistent with observations from other biomes showing that *Mesorhizobium* species exhibit high diversity in their tolerance to various pH conditions [51–53].

In the present study, the effectiveness of indigenous 30 *Mesorhizobium* strains in symbiosis association with Yezin-4 Myanmar chickpea variety by using sterilized vermiculite with MHN as a growth media. When the nitrogen fixation in terms of ARA per plant with the test *Mesorhizobium* strains were compared, four *Mesorhizobium* strains designated as *M. muleiense* MgnCP6, *M. plurifurium* MinCP15, *M.* sp. SalCP17, and *M. tianshanense* SalCP19 were found to be more effective than other tested strains. The effectiveness of a strain of rhizobia is due to the genetic interaction with the host plant, which is known as host-strain specificity. Thus, the selection of strains of rhizobia for cultivated legume varieties is a critical step in the production of the legume inoculants [54]. Strain of rhizobia which is effective on one legume cultivated may not be highly effective on others legume crops [5], chickpea in Than et al., [9] and soybean in Soe et al., [55]. It was observed that the inoculation of *M. tianshanense* SalCP19 gave significantly higher than *M.* sp. DemCP4 in nodule dry weight in Yezin-4 but *M.* sp. SalCP17 in Yezin-6.

When the nodule dry weight of the plants inoculated with test *Mesorhizobium tianshanense* SalCP19 in Yezin-4 chickpea variety was about 73% higher than that of *Mesorhizobium* sp. SalCP17 in Yezin-6 chickpea. Nodule dry weight of Yezin-4 chickpea variety was significantly higher than those of Yeizn-6 chickpea. Therefore, based on the results of nodulation efficiency of nodule dry weight of the plants Yezin-4 could be used in future experiment as superior host genotypes for high nitrogen fixation. Symbiotic nitrogen fixation depends on interactions among the genotype of the host plant, rhizobial strain genotype, and environment. In grain legume species, genotypic variability affected nodule number or mass or nitrogenase activity [56,57] pointed out that, through the use of plant genotypes in symbiotic ability, it is possible to identify genes responsible for a particular part of the process, depending on a particular rhizobial strain used.

The present study was observed that *M. tianshanense SalCP19* was significantly higher than *M.* sp. DemCP4 in ARA per plant and nodule dry weight of Yezin-4 chickpea variety. However, the *M.* sp SalCP17 was significantly better than *M.* sp. DemCP4 in the same parameters as above in the Yezin-6 chickpea variety, respectively. These two effective *Mesorhizobium* strains were collected from Salingyi Sagaing Region of a soil pH 7.27 with Meadow and Meadow Alluvial Soil. This finding is proved by [58] Mesorhizobial isolates originally from alkaline soils have been reported to be more effective and chickpea (*Cicer arietinum* L.) is a successful legume on alkaline soils [59].

The selection of effective rhizobial strains for cultivated legumes is a critical step in the production of high quality legume inoculant [60]. An essential desired characteristic for inoculum strains of root nodule bacteria is highly effective nitrogen-fixation with the intended host species [61]. In Tunisia, the field trials was showed that the inoculation of different chickpea cultivars with highly effective rhizobia strains seems to result in an increased number of nodules and shoot dry weight [62]. In our present study showed that inoculation of different chickpea cultivars with highly effective Mesorhizobia strains seems to result in an increased nodule and shoot dry weight of studied individual chickpea varieties.

All of these experiments were conducted under the control conditions by growing the plants in the sterilized vermiculite with MHN solution. So, the selected mesorhizobial strains and Myanmar chickpea cultivars should be evaluated in the field condition. Although the current experiments were a preliminary study, it could help for the future study for the inoculants production. We do hope that Myanmar *Mesorhizobium* strains will be able to use as Biofertilizer for chickpea cultivars that enhance crop production through nitrogen fixation and yield.

## 5. Conclusions

In conclusion, this is the first report finding of *M. muliense, M. tianshanense,* and *M. plurifurium* were identified from soil samples collected from major chickpea growing areas in Myanmar. *Mesorhizobium tianshanense* SalCP19 and *Mesorhizobium* sp. SalCP17 strains might be considered for rhizobial inoculants to use as Biofertilizer for chickpea production through nitrogen fixation in Myanmar.

**Author Contributions:** Conceptualization, K.M.S., T.Y., A.T.; methodology and investigation, K.M.S., A.Z.H., K.M., data organization and formal analysis, K.M.S.; writing—original draft preparation, K.M.S.; writing—review and editing, T.Y. and A.T.; supervision, T.Y. All authors have read and agreed to the published version of the manuscrip.

**Funding:** This research was supported by Japan Society for the Promotion of Science (P18079).

**Acknowledgments:** The authors are thankful to Japan Society for the Promotion of Science (JSPS) for their financial support of the present study. We are also very grateful to the members of Land Use Division, Department of Agriculture and members of Department of Agricultural Research, Ministry of Agriculture, Livestocks and Irrigation, Myanmar who helped for collecting soil samples and chickpea seeds for these experiments.

**Conflicts of Interest:** The authors declare no conflict of interest.

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
