# Peer review of "Diversity and Effectivity of Indigenous Mesorhizobium Strains for Chickpea (Cicer arietinum L.) in Myanmar"

_agronomy, doi:10.3390/agronomy10020287_

Round 1
Reviewer 1 Report
Area of strength: new knowledge about the isolated Mesorhizobium strains as well as their efficacy.
Area of weakness: it is necessary (in future) to determine the effectiveness of legume-bacteria symbiosis in field conditions.

Author Response
Agronomy Journal
Reviewer 1 comments to the Author
Thank you very much for all respective Reviewers comments and really appreciated for all your kind effort to our manuscript effectively.
We changed revised version with Red color according to the Reviewers comments (please see the pdf file).
|
1) Abstract exceeds 200 words recommended in journal guidelines |
Answer: Thank you so much indeed for your kind teaching. Corrected. Answer: Abstract: red color revised. Please see in lines 16 and 29. |
|
2) Line 21, 39, 175, 176, 195, 276, 283: typing errors |
Answer: Thanks indeed again for your kind comment. Revised accordingly with red color.
|
|
3) Line 59 TAL: - state what the abbreviation means |
Answer: TAL is nominated the strains that collected from Niftal Project. Department of Agricultural Research (DAR), Myanmar was closely cooperated with Niftal project in those cited year. They got TAL strains and extensively used. Currently, they are working with Dr. Davide Herridge, ACIAR project in DAR, Myanmar. Note: Revised version of current manuscript, we already deleted this TAL sentence according to your comment, “Generally introduction: line 75-82 and 57-63 it is necessary to fuse these parts because in this way it seems confusing to the readers”. |
|
4) all manuscript, especially in: Title and line 21, 25, 51, 110, 167, 219, 283, 284, 354: Apply the Codes for writing of Latin names |
Answer: Revised accordingly with Red color. Please forgive me about the pdf file. It was some changed in pdf files (the Codes for writing of Latin names). Word file is fine for us. Thank you very much for your kind comments. |
|
5) Line 29, 174: improve english translation |
Answer: Corrected. Please see the Line 152. |
|
6) Line 58 and line 76-77: repetition of the sentence. |
Answer: Deleted Line 58. Thank you very much for your kind comment. |
|
7) Generally introduction: line 75-82 and 57-63 it is necessary to fuse these parts because in this way it seems confusing to the readers |
Answer: Revised accordingly. Please see the Line 45-53. |
|
8) Line 89: there is no need to explain the pH methodology because the authors clarify this in the next section |
Answer: Deleted this section. |
|
9) Line 101: how many soil samples are collected for analysis |
Answer: Edited. Total six collected soil samples from major chickpea growing areas of Myanmar were analyzed. Please see the Line 74.
|
|
11) Line 113: specify the recipe of modified Hoagland solutions |
Answer: Added the recipe of modified Hoagland solutions. Please see the Line 86-91. |
|
13) Line 121: it is not clear how many pots are used in this stage of experiment? And how many seeds are applied per pots |
Answer: Edited. Six culture pots used in this stage of experiment. Six sterilized seeds were planted per pot. Please see the Line 99 and 100. |
|
14) Line 131: suspension is made? Nodule in tubes, any chemicals? |
Answer: Could I please I would like to respectfully answer your kind comment. The five nodules (≥ 2 mm in diameter) were collected and surface sterilization, the nodules were soaked in 70% ethanol for 3 min, 2.5% sodium hypochlorite (NaClO) solution for 15 min and washed five times with 0.9% autoclaved sodium chloride (NaCl) solution. The surface sterilized nodules were transferred separately into the autoclaved small test tubes and crushed. After crushing, we got some suspension. For every sample, a loopful of the suspension was streaked on Yeast Extract Mannitol Agar (YMA) plates. We did not use any other chemicals. |
|
15) Line 175: how many seeds per pot are used? |
Answer: Added. Six seeds per pot were used in our experiments. Please see the Line 159. |
|
16) Line 185: improve cited of reference according of journal guidelines |
Answer: Deleted according of journal guidelines. |
|
17) Line 195, 210: Bradyrhizobium or Mesorhizobium? |
Answer: Please forgive us about our mistake. This is Mesorhizobium. Thank you very much for your kind comments. Please see the Line 177-194. |
|
18) Line 205: which statistic program is used for analysis? |
Answer: Added statistic program. Please see the Line 186-189. |
|
19) Line 209: without a year |
Answer: Deleted and revised accordingly. |
|
20) Line 207: it is not clear is it isolated strains are Bradyrhizobium or Mesorhizobium? fast or slow growing? Information in text and table are not same!!! |
Answer: Please forgive us about our mistake. The 30 isolated strains are Mesorhizobium and fast-growers. Corrected 3-5 days (fast-grower) in text. Please see the Line 195 and 572. |
|
21) Generally Results: it is unnecessary to repeat the same tables and graphs in main text |
Answer: Deleted and revised some table and figure in main text. Please see the Line 217. |
|
22) Line 314, -321 unnecessarily, repeatedly stated |
Answer: Deleted the Line 314-321. |
|
23) Generally Discussion: properly discuss the statistical results of the experiment.
|
Answer: Revised accordingly. |
|
24) Line 347-355: point results of othere researches |
Answer: Revised accordingly. Please see the Line 333-336. |
|
25) Line 364-368: instead of this sentences authors should say something about future research |
Answer: Revised future research plan according to the comments. Please see the Line 349-354. |
|
26) Generally References must be redone according journals guidelines |
Answer: Revised according journal guidelines. Please see the Line 384-417 and 504-528. |

Reviewer 2 Report
The aim of this study was to evaluate the diversity and effectiveness of several Mesorhizobium strains for chickpea in Myammar. In general, the article is well written but there are some "gaps" that need to be addressed. I provide below a few suggestions that, if the authors decide to implement the paper will improved.
Comments
Title: Revise as “Diversity and effectiveness of indigenous Mesorhizobium strains for chickpea (Cicer arietinum L.) in Myammar
Abstract: This section is well written but should be revised since according to instructions for authors “the abstract should be a total of about 200 words maximum”.
Introduction: this section is well written.
Line 59: Explain abbreviation TAL
Line 64: Delete 2010. The authors should follow the instructions for authors for references. (Author 1, A.B.; Author 2, C.D. Title of the article. Abbreviated Journal Name Year, Volume, page range. e.g. volume number should be written in italics.
Material and Methods: This section is well written.
Subsections 2.7, 2.8: Information about the period that the pot experiment conducted should be added.
Subsection 2.8: It is not obvious if a control treatment (non-inoculant treatment) was included as a treatment in this experiment.
Line 166: delete Cicer arietinum.
Line 185: delete 1991.
Table 1 and 2 should be combined as one table.
Figure 1. This figure should be revised. Authors should not present information that already presented in table 1. Analysis of figure should be improved.
Results
Line209: delete 1994.
Table 3: Authors should add information about the regions that Mesorhizobium strains were isolated.
Lines 237-238: figure 2 and Table 3 should be replaced with table 4.
Figure 3 should be improved. Also the legend of this figure is missing from pdf version. Also, authors should include one column about control treatment.
Subsection 3.3: Authors should include data about control treatment. If this treatment was not included in this pot experiment authors should explain why this choice did.
Line 264: replace “rood” with “root”.
Discussion: This section should be slightly revised.
Line 287: correct as “a soil pH”
Line 288: correct as “a soil pH”
Line 305: delete “on the other hand”
Line 331: delete 2010 and 2013.
Lines 332-333: The phrase “In this study, ……… were investigated” should be deleted since already mentioned in other section of this manuscript.
Lines 336-339: Authors present again data that presented in results section. This part should be revised and authors should compare the treatments presenting information about the % differences among the strains.
Lines 347-350: This phrase should be revised. The part of the phrase “In the present study… MHN solution” should be deleted since already mentioned in other section of this manuscript.
Lines 261-263: Revise this phrase since there were no differences about dry shoot weight among the strains.
Author Response
Agronomy Journal
Reviewer 2 comments to the Author
Thank you very much for all respective Reviewers comments and really appreciated for all your kind effort to our manuscript effectively.
We changed revised version with Red color according to the Reviewers comments (please see the pdf file).
|
1Title: Revise as “Diversity and effectiveness of indigenous Mesorhizobium strains for chickpea (Cicer arietinum L.) in Myammar |
Answer: Thank you so much indeed for your kind teaching. Corrected. Please see in lines 1and 2. |
|
2) Abstract: This section is well written but should be revised since according to instructions for authors “the abstract should be a total of about 200 words maximum”. |
Answer: Abstract: red color revised. Please see in lines 16 and 29.
|
|
3) Line 59: Explain abbreviation TAL. |
Answer: TAL is nominated the strains that collected from Niftal Project. Department of Agricultural Research (DAR), Myanmar was closely cooperated with Niftal project in those cited year. They got TAL strains and extensively used. Currently, they are working with Dr. Davide Herridge, ACIAR project in DAR, Myanmar. Note: Deleted TAL sentence according to Reviewer 1 comment. |
|
4) Line 64: Delete 2010. |
Answer: Deleted accordingly. Please see in lines 54.
|
|
5) Subsections 2.7, 2.8: Information about the period that the pot experiment conducted should be added. |
Answer: Revised accordingly. Please see in lines 157 and 158 in subsection 2.6 (subsection 2.7 changed 2.6 according to Reviewer 1 comment). Please see in lines 181 and 183 in subsection 2.7 (subsection 2.8 changed 2.7 according to Reviewer 1 comment). Note: The period of these two experiments were a bit far from each other. This is because we studied 4 leguminous crops: chickpea, green gram, black gram and groundnut. It took time for doing experiments crop by crop continuously. |
|
6) Subsection 2.8: It is not obvious if a control treatment (non-inoculant treatment) was included as a treatment in this experiment. |
Answer: Thank you very much for your kind comments. The inoculation and growing condition of pot experiment was also conducted as the above experiment in Subsection 2.7. Please see in Subsection 2.6, I mentioned that the non-inoculated treatment, a control treatment was also provided. On the other hand, I also mentioned in Table 5 that NDW and ARA of uninoculated treatment is zero for both chickpea varieties, RDW of unioculated treatment is 0004 g plant-1 for Yezin-4 and 0.05 g plant-1 for Yezin-6 chickpea variety; and SDW of unioculated treatment is 0.04 g plant-1 for Yezin-4 and 0.11 g plant-1 for Yezin-6 chickpea variety. Note: subsection 2.8 changed to 2.7 according to Reviewer 1 comment. |
|
7) Line 166: delete Cicer arietinum. |
Answer: Deleted accordingly. Please see in Line 144. |
|
8) Line 185: delete 1991. |
Answer: Deleted. Now see in Line 164.
|
|
9) Table 1 and 2 should be combined as one table. |
Answer: Yes, respective reviewer. We also wanted to combine as one table but this table was not fit with paper layout margin. So, we finally divided into two tables. Thank you very much for your kind comments. I hope you would kindly give approval for that. |
|
10) Figure 1. This figure should be revised. Authors should not present information that already presented in table 1. Analysis of figure should be improved. |
Answer: Yes, you are right. For Figure 1, we aimed to mention for all the readers to easily imagine where the collected soil samples areas in Myanmar. If you dislike for that, will we delete the Figure 1 or Table 1? |
|
11) Line 209: delete 1994. |
Answer: Deleted. Please see in Line 164. |
|
12) Table 3: Authors should add information about the regions that Mesorhizobium strains were isolated. |
Answer: Revised accordingly. Please see in Table 3. |
|
13) Lines 237-238: figure 2 and Table 3 should be replaced with table 4. |
Answer: Replaced. Please see in Line 230.
|
|
14) Figure 3 should be improved. Also the legend of this figure is missing from pdf version. Also, authors should include one column about control treatment. |
Answer: Improved legend. Could I please I would like to respectfully want to explain about our research nature. We isolated 30 strains from collected soil samples from Myanmar. For the screening of effective for nitrogen fixation experiment, we would like to choose only the highest nitrogen fixation strains compare with 30 strains individually. In which, the control treatment plants did not have any nodule and could not measure nitrogen fixation as well. So, we did not add control data on the screening experiment. Only compared individual strains because we would like to get the highest nitrogen fixation strains for future experiment. Thank you very much for your kind consideration. We hope you would kindly give approval for that. |
|
15) Subsection 3.3: Authors should include data about control treatment. If this treatment was not included in this pot experiment authors should explain why this choice did. |
Answer: Could I please I would like to respectfully want to answer you the following. In Table 5, we mentioned Table 5 Legend as “NDW and ARA of uninoculated treatment is zero for both chickpea varieties, RDW of unioculated treatment is 0.04 g plant-1 for Yezin-4 and 0.05 g plant-1 for Yezin-6 chickpea variety; and SDW of unioculated treatment is 0.04 g plant-1 for Yezin-4 and 0.11 g plant-1 for Yezin-6 chickpea variety”. Normally, we used the control treatment in our experiments was that only checking for nodulation or checking the contamination. If we added the control treatment data in this Table, NDW and ARA were zero. So, the same results appeared after analysis of statistic. So, we didn’t mention inside the table and could not compare with the control by using statistic. Please kindly consider for that. |
|
16) Line 264: replace “rood” with “root”. |
Answer: Deleted. Please see in Line 253. |
|
17) Discussion: This section should be slightly revised. |
Answer: Slightly revised accordingly. Please see the Discussion. |
|
18) Line 287: correct as “a soil pH” |
Answer: Corrected. Please see in Line 277. |
|
19 Line 288: correct as “a soil pH” |
Answer: Corrected. Please see in Line 277. |
|
20) Line 305: delete “on the other hand” |
Answer: Deleted. Please see in Line 294. |
|
21) Line 331: delete 2010 and 2013. |
Answer: Deleted. Please see in Line 312. |
|
22) Lines 332-333: The phrase “In this study, ……… were investigated” should be deleted since already mentioned in other section of this manuscript. |
Answer: Deleted accordingly. |
|
23) Lines 336-339: Authors present again data that presented in results section. This part should be revised and authors should compare the treatments presenting information about the % differences among the strains.
|
Answer: Revised. Please see in Line 315-317. |
|
24 Lines 347-350: This phrase should be revised. The part of the phrase “In the present study… MHN solution” should be deleted since already mentioned in other section of this manuscript. |
Answer: Revised. Please see in Line 325 and 328. |
|
25) Lines 261-263: Revise this phrase since there were no differences about dry shoot weight among the strains. |
Answer: Revised. Please see in Line 251 and 252. |
